# Characterization of a New Glucose-Tolerant GH1 β-Glycosidase from *Aspergillus fumigatus* with Transglycosylation Activity

**DOI:** 10.3390/ijms24054489

**Published:** 2023-02-24

**Authors:** Lucas Matheus Soares Pereira, Aline Vianna Bernardi, Luis Eduardo Gerolamo, Wellington Ramos Pedersoli, Cláudia Batista Carraro, Roberto do Nascimento Silva, Sergio Akira Uyemura, Taísa Magnani Dinamarco

**Affiliations:** 1Department of Chemistry, Faculty of Philosophy, Sciences and Literature of Ribeirão Preto, University of São Paulo, Ribeirão Preto 14040-901, SP, Brazil; 2Department of Biochemistry and Immunology, Ribeirão Preto Medical School, University of São Paulo, Ribeirão Preto 14049-900, SP, Brazil; 3Department of Clinical, Toxicological and Bromatological Analysis, School of Pharmaceutical Sciences of Ribeirão Preto, University of São Paulo, Ribeirão Preto 14040-903, SP, Brazil

**Keywords:** β-glycosidase, glucose stimulation, transglycosylation activity, enzymatic hydrolysis

## Abstract

Concern over environmental impacts has spurred many efforts to replace fossil fuels with biofuels such as ethanol. However, for this to be possible, it is necessary to invest in other production technologies, such as second generation (2G) ethanol, in order to raise the levels of this product and meet the growing demand. Currently, this type of production is not yet economically feasible, due to the high costs of the enzyme cocktails used in saccharification stage of lignocellulosic biomass. In order to optimize these cocktails, the search for enzymes with superior activities has been the goal of several research groups. For this end, we have characterized the new β-glycosidase AfBgl1.3 from *A. fumigatus* after expression and purification in *Pichia pastoris* X-33. Structural analysis by circular dichroism revealed that increasing temperature destructured the enzyme; the apparent T_m_ value was 48.5 °C. The percentages of α-helix (36.3%) and β-sheet (12.4%) secondary structures at 25 °C were predicted. Biochemical characterization suggested that the optimal conditions for AfBgl1.3 were pH 6.0 and temperature of 40 °C. At 30 and 40 °C, the enzyme was stable and retained about 90% and 50% of its activity, respectively, after pre-incubation for 24 h. In addition, the enzyme was highly stable at pH between 5 and 8, retaining over 65% of its activity after pre-incubation for 48 h. AfBgl1.3 co-stimulation with 50–250 mM glucose enhanced its specific activity by 1.4-fold and revealed its high tolerance to glucose (IC_50_ = 2042 mM). The enzyme was active toward the substrates salicin (495.0 ± 49.0 U mg^−1^), pNPG (340.5 ± 18.6 U mg^−1^), cellobiose (89.3 ± 5.1 U mg^−1^), and lactose (45.1 ± 0.5 U mg^−1^), so it had broad specificity. The V_max_ values were 656.0 ± 17.5, 706.5 ± 23.8, and 132.6 ± 7.1 U mg^−1^ toward *p*-nitrophenyl-β-D-glucopyranoside (pNPG), D-(-)-salicin, and cellobiose, respectively. AfBgl1.3 displayed transglycosylation activity, forming cellotriose from cellobiose. The addition of AfBgl1.3 as a supplement at 0.9 FPU/g of cocktail Celluclast^®^ 1.5L increased carboxymethyl cellulose (CMC) conversion to reducing sugars (g L^−1^) by about 26% after 12 h. Moreover, AfBgl1.3 acted synergistically with other *Aspergillus fumigatus* cellulases already characterized by our research group—CMC and sugarcane delignified bagasse were degraded, releasing more reducing sugars compared to the control. These results are important in the search for new cellulases and in the optimization of enzyme cocktails for saccharification.

## 1. Introduction

Plant cell wall (PCW), which consists of cellulose fibers embedded within a matrix of hemicelluloses, pectin, and lignin [1], represents the most abundant renewable resource on Earth [2,3]. Because PCW is rich in polysaccharides, it can be employed as feedstock to obtain monomers such as glucose (C6) and xylose (C5) through several bioprocesses [4]. To this end, pretreatment (which makes cellulose more available to enzymes) and enzymatic hydrolysis are combined to ensure that the biomass is efficiently used [5,6].

Considering enzymatic hydrolysis, commercial cocktails for hydrolyzing lignocellulosic biomass are composed mainly of cellulases, which can hydrolyze cellulose polymers.

These enzymes work synergistically to break down polysaccharides and crystalline cellulose [7]. First, endoglucanases (EGL, EC 3.2.1.4) cleave β-1,4-glycosidic bonds in amorphous regions of the cellulose chains, reducing end-acting cellobiohydrolases (EC 3.2.1.176) and non-reducing end-acting cellobiohydrolases (EC 3.2.1.91) release cellooligosaccharide units like cellobiose. Then, β-glycosidases (BG; EC 3.2.1.21) use cellobiose and low-molecular-weight oligomers as substrates and cleave them into glucose for further fermentation [8,9].

Enzyme cocktails are expensive. Therefore, optimizing the hydrolysis step of lignocellulose biomass conversion to fermentable sugars is crucial for some biorefineries [10]. Lignocellulolytic enzyme cocktails can be improved by different strategies; finding more efficient enzymes is one of them [11].

β-glycosidases are dual-character enzymes that catalyze both the synthesis and degradation of glycosidic bonds, so they are good candidates for biorefineries. In recent years, high initial biomass concentration has been established as an essential condition for developing an economically viable process to produce lignocellulosic ethanol.

Under these hydrolysis conditions, increased concentration of fermentable sugars in hydrolysates is expected to reduce the number of operation steps and costs [12,13]. Nevertheless, to date, no efficient processes for biomass hydrolysis have been developed under these conditions, partly because commercially available enzyme preparations are inefficient. In fact, the presence of high concentrations of glucose, xylose, or both in hydrolysates intensifies product inhibition, decreasing the efficiency of enzymatic hydrolysis [13]. Glucose and xylose concentrations in these hydrolysates are estimated to reach 650–1000 and 300–400 mmol L^−1^, respectively [14,15,16,17].

To solve this problem, some alternatives such as developing simultaneous saccharification and fermentation processes, constructing bioreactors for continuous glucose removal, and new strategies for feeding bioreactors with enzymes, substrates, or both have been proposed [13,18]. Using product-tolerant enzymes may also represent a promising approach.

Some β-glycosidases, most of them belonging to family 1 of glycosyl hydrolases (GH1), have been reported to be highly tolerant to glucose; in some cases, such tolerance is accompanied by a stimulatory effect [19]. Interest in studying these enzymes is great particularly because they may be applied in 2G ethanol production. Indeed, tolerance to reaction products reduces the need for replacing these enzymes in bioreactors, whereas most β-glycosidases are inhibited by even low glucose concentrations [20,21,22].

Native enzymes are often unsuitable for large-scale applications, and their catalytic properties need to be improved in terms of stereoselectivity, chemoselectivity, and process-related aspects, including stability under certain pH and temperature conditions and good activity in the presence of high concentrations of the substrate, potential inhibitors, or both [20,23]. Here, we present the cloning, heterologous expression in *Pichia pastoris*, and biochemical characterization of a glycoside hydrolase family 1 protein, named AfBgl1.3, from *Aspergillus fumigatus*. We will show that AfBgl1.3 is insensitive to glucose inhibition and displays relatively high transglycosylation activity.

## 2. Results and Discussion

### 2.1. AfBgl1.3 Structural Prediction

The AfBgl1.3 gene from *A. fumigatus* (Afu1g14710, Gene ID 3509863, GenBank access number XM_747747.1) is a 1449 bp open reading frame (ORF) encoding a β-glycosidase (AfBgl1.3) with 483 amino acid residues and a theoretical molecular weight of 54.0 kDa (http://www.aspergillusgenome.org/, accessed on 20 October 2019; https://www.uniprot.org/uniprot/Q4WRY0, accessed on 15 February 2020). On the basis of sequence similarity and according to the CAZy database, we classified recombinant AfBgl1.3 into GH1.

By using the program SignalP 5.0, we did not identify any peptide signal at the N-terminal even though this protein was expressed and secreted by the fungus *A. fumigatus* [24]. The software NetNGlyc 1.0 predicted one *N*-glycosylation site at position N385. *N*-glycosylation of asparagine residues is an essential post-translational modification for protein folding, secretion, stability, and other enzymatic properties [25]. Furthermore, through modeling, we did not identify any cysteine residues involved in disulfide bonds in AfBgl1.3.

Multiple sequence alignment of AfBgl1.3 (Uniprot ID: Q4WRY0) with homologous sequences of GH1β-glycosidases from other species, such as *Oryza sativa* Os4BGlu18 (Uniprot ID:Q7XSK0, PDB:7D6B), *Trichoderma reesei* TrBgl2 (Uniprot ID: O93785, PDB: 3AHY), *Trichoderma harzianum* ThBgl2 (Uniprot ID: A0A0F9XM91, PDB: 5JBO), and *Jeotgalibacillus malaysiensis* BglD5 (Uniprot ID: A0A0B5ARU7), revealed percent identity of 40%, 50%, 68.6%, and 43%, respectively (Figure 1) [26,27,28,29].

A conserved region in the N-terminal motif comprising 15 amino acids has been described. The sequence of this region is given by: [FX-(F/Y/W/M)-(G/S/T/A)-X-(G/S/T/A)-X-(G/S/T/A)-(G/S/T/A)-(F/Y/N/H)-(N/Q)-XEX-(G/S/T/A)] (where X is described as a variable amino acid in the sequence motifs). In AfBgl1.3, we identified this sequence as FLWGFATASYQIEGA and the TXNEP and I/VTENG motifs, which contain the catalytic glutamate residues (E173 and E384) [26,30].

### 2.2. Expression of Recombinant β-Glycosidase, AfBgl1.3, in P. pastoris and Its Characterization

We expressed recombinant AfBgl1.3 in *P. pastoris* X-33. After purification by Ni Sepharose 6 Fast Flow resin chromatography, AfBgl1.3 showed a single protein band on SDS-PAGE (Figure 2) and yielded about 6.4 mg of AfBgl1.3/L of culture.

SDS-PAGE revealed a considerably higher molecular mass than the calculated molecular mass (54.0 kDa). This was expected due to hyperglycosylation activities in proteins expressed by *P. pastoris* and addition of 6x-His-tag. Nevertheless, one potential *N*-glycosylation site was predicted at residue N385, which is conserved in other GH1 β-glycosidases. Treatment with Endo H (New England Biolabs, Ipswich, MA, USA) did not reduce the AfBgl1.3 molecular mass because one site is not enough to reduce a molecular mass that can be seen on SDS-PAGE.

We modeled the three-dimensional structure of AfBgl1.3 as predicted by AlphaFold; the catalytic residues GLU173 and GLU384 are indicated with black lines and the N-terminal motif is highlighted in green (Figure 3A).

We analyzed the secondary structure by Far-UV CD spectrum (CD) at 25 °C (Figure 3B), which allowed us to estimate 36.3% α-helices and 12.4% β-sheets by BeStSel [31]. These results resembled the results achieved with Phyre2 (35.0% and 14.0%) and the Kabsch and Sander method (39.8% and 15.3%), which indicated that AfBgl1.3 may be in its native conformation under the CD analysis conditions [32,33]. Furthermore, the enzyme TrBgl2 (PDB:3AHY), which shares 69% identity with AfBgl1.3, has a similar proportion of α-helices (41.3%) and β-sheets (15.5%) [26].

We also evaluated structural changes by subjecting AfBgl1.3 to different temperatures (25–80 °C) for 30 min, as shown in Figure 3C. Upon increasing temperature, the spectroscopic profile changed: the maximum peak at 190–195 nm decreased progressively. The α-helix proportion of the negative peak at 208 and 222 nm decreased. In addition, the 200-nm shifted, especially going from 40 to 50 °C, indicating protein denaturation [34].

On the basis of the CD results, the melting temperature (Tm) was 48.5 °C (Figure 3D). For comparison, literature values based on CD are Tm of 49 °C for GH1ThBgl (40 mM HEPES buffer pH 7.5 and 150 mM NaCl), 45.2 °C for GH1 Gluc1C (50 mM sodium phosphate buffer pH 7.5), and 64.5 °C for GH3 Bgl7226 (5 mM Tris-HCl buffer, pH 7.0) [35,36,37].

We expected the lower Tm of AfBgl1.3 because the absence of disulfide bonds in its structure makes it less resistant to conformational changes. The presence of disulfide bonds, mainly at the N or C terminus or in the α-helix regions, is one of the major factors underlying increased thermal stability [38]. Table 1 shows the proportion of the secondary structures in recombinant AfBgl1.3 at different temperatures. At higher temperatures, the percentage of α-helix and antiparallel β-sheet conformation decreased and increased, respectively.

The presence of 22 tryptophan residues made AfBgl1.3 more sensitive to Intrinsic Tryptophan Fluorescence Emission (ITFE) analysis (Figure 3E). During ITFE analysis, increasing the temperature from 25 to 40 °C did not cause significant changes, showing that the polarity of the tryptophan residues did not change, either. On the other hand, a red shift was evident upon increasing the temperature from 40 to 50 °C, indicating transition and greater exposure of the tryptophan residues to the polar solvent, which decreased excitation, as evidenced by the lower energy of the fluorescence emission; the maximum wavelength (λ_max_) shifted from 335 to 340 nm. At 70 and 80 °C, there were no marked shifts, but the intensities decreased due to higher exposure. On the basis of the ITFE data, we determined Tm as 49.3 °C (Figure 3F), as previously indicated by the CD technique.

### 2.3. Substrate Specificity

According to substrate specificity, β-glycosidases can be classified into: (i) aryl-β-glycosidases, which have high affinity for aryl-β-glycoside (D-(-)-salicin and *p*-nitrophenyl-β-D-glucopyranoside); (ii) cellobiases, which act only on oligosaccharides (cellobiose); and (iii) β-glycosidases of broad specificity, which act on different types of substrates, the most common group being observed in β-glycosidases [39].

We tested the AfBgl1.3 activity toward all these substrates and observed that the enzyme acts on all of them. We observed greater activity toward D-(-)-salicin (495.0 ± 49.0 U/mg) and pNPG (340.5 ± 18.6 U/mg) as compared to cellobiose (89.3 ± 5.1 U/mg) and lactose (45.1 ± 0.5 U/mg). Therefore, recombinant AfBgl1.3 can be classified as a β-glycosidase of broad specificity (Figure 4).

### 2.4. Effect of pH and Temperature on AfBgl1 Activity and Stability

We characterized the pH and temperature profiles and the thermal stability of purified AfBgl1.3 toward the substrate D-(-)-salicin. The enzyme was active at pH ranging from 5.4 to 6.4 (95–100% of maximum activity); the highest activity occurred at pH 6.0 (Figure 5A).

pH is a critical factor that affects enzymatic activity. AfBgl1.3 was considerably stable at pH ranging from 5.0 to 8.0, retaining about 65% of its original activity after pre-incubation for 48 h. At more acidic pH (pH 3.0 and 4.0), AfBgl1.3 completely lost its activity (Figure 5C). The activities of β-glycosidases from different fungal species peak in acidic medium, between pH 4.0 and 6.0 [40,41,42]. β-glycosidases from *F. chlamydosporum* HML278, *Penicillium chrysogenum*, *Penicillium citrinum* UFV1, and *Aspergillus niger* have maximal activities at pH 5.0 [40,43,44].

AfBgl1.3 showed excellent activity at 40 °C (100%), maintaining about 90% and 95% of its maximum activity at 35 and 45 °C, respectively (Figure 5B). This characteristic is also desirable for industrial processes such as food and lignocellulose biorefineries [45]. Moreover, at 30 and 40 °C, AfBgl1.3 retained 90% and 50% of its initial activity, respectively, after incubation for 24 h. At 45 °C, AfBgl1.3 retained 70% of its activity after pre-incubation for 1 h and completely lost its activity after 15 h. At 50 °C, AfBgl1.3 did not show any activity (Figure 5D). These results corroborate the findings obtained from CD and ITFE analyses, which showed Tm values of 48.5 and 49.3 °C, respectively.

AfBgl1.3 was active at higher temperature, but stability at 30 and 40 °C is one of the most desired characteristics of enzymes intended for industrial processes (simultaneous saccharification and fermentation and consolidated bioprocessing) because the enzymes would work under the same conditions as fermenting microorganisms, such as *Saccharomyces cerevisiae* [46,47].

Another relevant characteristic of AfBgl1.3 was its storage stability at 4 °C. AfBgl1.3 retained about 75% of its activity after 140 days when it was stored in 20 mM sodium phosphate and 500 mM NaCl (pH 7.4) buffer, demonstrating good performance for industrial purposes (Figure 5E).

Table 2 shows the biochemical properties of some β-glycosidases. Almost all the enzymes are not stable for a long time at different temperatures. Although BglD5 and bgl1 are active at 65 and 70 °C, respectively, they are not stable at these temperatures for more than 60 min. Compared to MaGlu1A (34% residual activity after 30 min), BglNB11 (40% residual activity after 60 min), and ThBGL1A (100% residual activity after 60 min), which are thermally stable at 40 °C, we can conclude that AfBgl1.3 was even more stable than these enzymes because it kept 60% of its activity after incubation for 15 h.

### 2.5. AfBgl1.3 Kinetic Parameters

We determined the AfBgl1.3 kinetic parameters (K_M_, V_max_, and *k_cat_*) for the substrates *p*-nitrophenyl-β-D-glucopyranoside, salicin, and cellobiose (Table 3). For pNPG, *k_cat_*/K_M_ was 10-fold higher (7.8 × 10^4^ M^−1^s^−1^) compared to cellobiose (7.8 × 10^3^ M^−1^s^−1^), which was consistent with the substrate preference and specificity proposed by Nam et al. (2010) [56]. These results resembled the results obtained for other β-glycosidases of the GH1 family. For example, β-glycosidase TsBGL from *Thermofilum* sp. ex4484_79 has about 58-fold higher *k_cat_*/K_M_ for pNPG (220.5 M^−1^s^−1^) compared to cellobiose (3.81 M^−1^s^−1^) [57]. Furthermore, β-BGLA glycosidase from *Alteromonas* sp. L82 has about 28-fold higher *k_cat_* /K_M_ (36.5 M^−1^S^−1^) for pNPG than cellobiose (1.3 M^−1^s^−1^) [58].

### 2.6. Effects of Additives on AfBgl1.3 Activity

Some metal ions can activate or inhibit enzymatic activity. Investigation into how metal ions affect AfBgl1.3 activity showed that Mg^2+^, Mn^2+^, Ca^2+^, K^+^, and NH_4_^+^ did not activate the enzyme significantly. In contrast, Fe^2+^ and Co^2+^ decreased the enzymatic activity to 31.4% and 39.9%, respectively, whereas Cu^2+^, Zn^2+^, and Ag^+^ dramatically inactivated the enzyme. Cu^2+^ can inhibit enzymatic activity by acting on thiol sites of the side chain of cysteine residues, which inhibits the enzyme after these functional groups are oxidized [59] (Table 4).

The effects of ions on other β-glycosidases have been demonstrated, and they vary. Addition of all the tested ions to Bgl from *Aureobasidium pullulans* does not affect its activity [60]. On the other hand, Cu^2+^ and Pb^2+^ inhibit *Penicillium pinophilum* Bgl [61,62], while *Sporidiobolus pararoseus* Bgl is inhibited only by Ag^+^ and Hg^2+^ and partially inhibited by Cu^2+^ and Zn^2+^ [63]. Finally, Na^+^, K^+^, Mg^2+^, and Ba^2+^ stimulate and Zn^2+^, Cu^2+^, Al^3+^, and Fe^3+^ inhibit Bgl from *Microbulbifer* sp. ALW1 (named MaGlu1A) [53].

Organic compounds can affect enzyme structure and cofactor functioning, directly impacting the enzymatic activity. Here, EDTA affected the AfBgl1.3 activity little (94.5%), suggesting that AfBgl1.3 is not a metalloprotein because EDTA binds to divalent ions [64,65]. A reducing agent, β-mercaptoethanol, which reduces disulfide bonds, did not significantly affect the AfBgl1.3 activity either (103.5%). In contrast, DTT, which is generally used to stabilize the sulfhydryl groups of proteins, stimulated the AfBgl1.3 activity to 111.2%. These findings indicated that AfBgl1.3 did not lose its activity under reducing conditions because it did not have disulfide bonds, which kept its structure intact in the presence of these reducing agents [54,66,67]. As for organic co-solvents, DMSO improved the AfBgl1.3 activity to 116%. DMSO is an organic solvent that is used for dissolving organic substances such as carbohydrates and polymers. Depending on DMSO concentration, it can increase enzyme stability and hence enzymatic activity [68,69].

The non-ionic detergents Tween 20 and Triton X-100 affected the AfBgl1.3 activity positively, increasing the activity by 118% and 123%, respectively. In contrast, the ionic detergents SDS and SLS strongly inhibited AfBgl1.3.

Another factor that prevents enzymes from being applied in the industry is their inhibition by glucose, which at high concentrations decreases the efficiency of enzymatic hydrolysis. Therefore, β-glycosidases tolerant and stimulated by glucose are desirable for enzyme cocktails because they can increase the concentration of fermentable sugars at the end of the process [70].

We investigated how glucose affected the AfBgl1.3 activity by inhibitory assay. Efficient cellulose hydrolysis generates high glucose concentration, so tolerance to glucose is an essential property of β-glycosidases. This is a natural mechanism of enzymatic inhibition by product, but some β-glycosidases are more resistant than others, which confers an advantage to bioprocesses on a large scale. β-glycosidases can be classified as (I) β-glycosidases that are strongly inhibited by low glucose concentrations, (II) β-glycosidases that tolerate glucose, (III) β-glycosidases that are stimulated by low glucose concentrations and inhibited at higher glucose concentrations, and (IV) β-glycosidases that are not inhibited by high glucose concentrations. Most β-glycosidases that are resistant to glucose belong to the family GH1, but some GH3 can also be found in this class [19].

We added increasing glucose concentrations (0–3000 mM) and measured the relative AfBgl1.3 activity (Figure 6). The AfBgl1.3 activity was strikingly stimulated by a factor of 1.4 in the presence of 250 mM glucose. This activity declined gradually, by 90%, 50% and 30%, at higher glucose concentrations of 1000, 2000, and 3000 mM, respectively. This tolerance corresponded to IC_50_ of 2042 mM. Thus, we classified AfBgl1.3 as β-glycosidase stimulated by low glucose concentrations and inhibited by higher glucose concentrations.

Stimulatory effects of glucose on characterized β-glycosidases have been described, and dosage-dependent effects have been observed on different enzymes [7,71,72,73,74,75]. At 1000 mM glucose, the relative activity of β-glucosidase from *Bacillus subtilis* RA10 is 70%, while AfBgl1.3 showed 90% activity. Furthermore, we observed higher IC50 for AfBgl1.3 (2042 mM) compared to the IC50 of Cel1A from *T. reesei* (650 mM) and HiBGL (800 mM). Thus, AfBgl1.3 has potential industrial application for efficient cellulose saccharification [76].

### 2.7. AfBgl1.3 Transglycosylation Activity

To investigate whether transglycosylation plays a role in tolerance to glucose, we incubated AfBgl1.3 with cellobiose, as substrate (1% *w*/*v*), in the presence of glucose, and we analyzed the samples by thin layer chromatography (TLC) to verify the transglycosylated product (Figure 7).

Figure 7A shows a comparison across 1% cellobiose added to 0, 10, 20, and 45 g L^−1^ glucose at 30 and 60 min. We selected the two periods because they represent the regular activity assay. We detected a transglycosylation product band in the TLC plate for all the conditions. To check the effect of glucose, we added 0, 10, 20, and 45 g L^−1^ as a substrate without cellobiose. Except glucose, we detected no sugar, which indicated that transglycosylation did not occur when only glucose was present (Figure 7B). The slight smears result from glucose overload. Analysis by High Performance Liquid Chromatography (HPLC) revealed that the reaction of AfBgl1.3 with cellobiose resulted in the transglycosylation product cellotriose (Figure 7C).

The formation of different products by this mechanism has been suggested to occur due to glucose being accommodated under different orientations at the +1 subsite of the enzyme, which allows β-1-2, β-1-3, or β-1-4 glycosidic bonds to be formed between glucose (acceptor) and the glycosyl-enzyme intermediate during the deglycosylation step [77].

Transglycosylation activity has been described in other glucose-stimulated and glucose-tolerant β-glycosidases. β-glycosidase Unbgl1A can form cellotriose from cellobiose after reaction at 37 °C for 48 h [75]. β-glycosidase Td2F2 generates cellobiose, sophorose, laminaribiose, and gentiobiose when it reacts with 1.25, 5.0, or 10 mM cellobiose in the presence of 125–1000 mM glucose. However, when Td2F2 is incubated with cellobiose only, the main product of the reaction is glucose, and no transglycosylation product is detected. This contrasts with AfBgl1.3, which can form a product in the presence of cellobiose without glucose addition [78].

Recently, it has been observed that the transglycosylation activity can affect the production of lignocellulolytic enzymes positively: some products originating by this mechanism, such as cellobiose and sophorose, induce the production of cellulases in fungi, such as *T. reesei*. This feature has been investigated and described as an economical way to produce more cellulases, which can reduce the cost involved in enzyme production [79,80]. Furthermore, the transglycosylation product cellotriose is interesting for the food industry as a potential prebiotic [81].

### 2.8. Effect of AfBgl1.3 on Saccharification

To evaluate the efficiency of supplementing the commercial enzyme cocktail Celluclast^®^ 1.5L with 2.5 μg of recombinant AfBgl1.3, during CMC breakdown we incubated the mixture under the conditions optimized for AfBgl1.3 (pH 6.0 and 40 °C) and monitored the released sugars for 12 h (Figure 8). Addition of recombinant AfBgl1.3 to the cocktail increased CMC saccharification (Figure 8A,B). After incubation for 12 h at 0.1 FPU/g of cocktail supplemented with 2.5 μg of β-glucosidase, the concentration of soluble sugars was around 0.90 g L^−1^. When we used 0.9 FPU/g of cocktail supplemented with 2.5 μg of AfBgl1.3, CMC conversion to reducing sugars increased by about 26% compared to the cocktail alone (Figure 8B).

Next, we associated AfBgl1.3 with cellulases from *A. fumigatus*, namely endoglucanase (Af-EGL7) and cellobihidrolase (AfCel6A), which had been previously characterized [82,83].

We observed a synergistic effect when we associated Af-EGL7, AfCel6A, and AfBgl1.3 at a 1:10:50 ratio, the ratio that provided the greatest release of reducing sugars during CMC hydrolysis (3.60 ± 0.07 mg mL^−1^). This release was 28.6% higher compared to the release of reducing sugars (2.80 ± 0.10 mg mL^−1^) during the control reaction in the presence of Af-EGL7 + AfCel6A (1:10) (Figure 9A).

When we associated AfBgl1.3 and Af-EGL7 (50:1), the release of reducing sugars was 616% higher compared to Af-EGL7 alone. Similarly, there was a 46% higher release of reducing sugars when we associated AfBgl1.3 and AfCel6A. Therefore, AfBgl1.3 acted in synergy with cellulases from *A. fumigatus* (Figure 9A).

We also evaluated the synergistic action of the three enzymes during sugarcane delignified bagasse (SDB) hydrolysis at 40 °C and 1000 rpm for 48 h. We used the optimal concentrations statistically estimated for the recombinant enzymes. Figure 9B shows that the presence of AfBgl1.3 in the reactions enhanced the release of reducing sugars by about 13%. AfBgl.3 prevented cellobiose from accumulating during the hydrolysis stage, thereby avoiding inhibition of other enzymes and resulting in slightly higher yields.

## 3. Materials and Methods

### 3.1. Strains, Culture Conditions, Vector and Materials

*A. fumigatus* Af293 was cultivated in Yeast Agar Glucose (YAG) solid medium (0.5% (*w*/*v*) yeast extract, 2% (*w*/*v*) dextrose, 1% (*w*/*v*) vitamin supplement, 1.8% (*w*/*v*) agar, and 0.1% (*v*/*v*) trace element solution) at 37 °C for 3 days. From this suspension, a pre-inoculum of 10^8^ conidia was prepared in 50 mL of Yeast Nitrogen Base (YNB) liquid medium (0.05% (*w*/*v)* yeast extract, 0.1% (*v*/*v*) trace element solution, and 5% (*v*/*v*) salt solution 20X) added with 1% (*w*/*v*) fructose, and the culture was incubated at 37 °C and 200 rpm for 16 h.

Next, the mycelia were washed and transferred to 50 mL of YNB liquid medium, added with 1% (*w*/*v*) SEB (exploded sugarcane bagasse), and incubated at 37 °C and 200 rpm for 24 h. Then, the mycelia were used for extracting RNA.

The plasmid pPICZαA (Invitrogen, Carlsbad, CA, USA) was used for cloning, sequencing, and expressing AfBgl1.3.

*E. coli* DH10β was used to propagate the recombinant pPICZαA expression vector (containing the Afu1g14710 gene sequence). To select positive clones, the bacteria were cultured in LB Low Salt solid medium (1.5% (*w*/*v*) agar, 1% (*w*/*v*) tryptone, 0.5% (*w*/*v*) NaCl, and 0.5% (*w*/*v*) yeast extract) containing 100 µg mL^−1^ zeocin at 37 °C for 16 h. *P. pastoris* X-33 was transformed with the recombinant pPICZαA expression vector and used for producing heterologous protein. The growth conditions were 200 rpm at 30 °C.

Cellobiose, *p*-nitrophenyl-β-D-glucopyranoside, D-(-)-salicin, and carboxymethyl cellulose (CMC) were purchased from Sigma (Sigma-Aldrich, St. Louis, MO, USA).

### 3.2. Sequence and Structural Analysis

Protein sequence was analyzed using bioinformatics tools, such as servers SignalP (http://www.cbs.dtu.dk/services/SignalP/, accessed 15 February 2020), for determining signal peptides, and NetNGlyc 1.0 (http://www.cbs.dtu.dk/services/NetNGlyc/, accessed 15 February 2020), for predicting possible glycosylation sites. AfBgl1.3 tridimensional structure was predicted using AlphaFold [84] through the cloud computing platform ColabFold v1.5.1 [85]. The amino acid FASTA sequence for the corresponding protein was retrieved from the NCBI database (https://www.ncbi.nlm.nih.gov/protein/, accessed on 16 February 2023), under the accession number XP_752840.1 and was submitted to ColabFold for model prediction with the application of AMBER force field for the relaxation of the firstly ranked modeled structure. The structure was built based on MMseqs2 homology search with 20 recycles. The best-ranked model was selected and submitted to the web server GalaxyRefine (https://galaxy.seoklab.org/cgi-bin/submit.cgi?type=REFINE, accessed on 15 February 2023) [86], for structure refinement with default parameters and the retrieved model was submitted for validation through PROCHECK [87,88]. The model was properly validated (Appendix A) and the structure was visualized and analyzed on BIOVIA Discovery Studio Visualizer v21.1.0.20298.

### 3.3. RNA Extraction, cDNA Synthesis, and Gene Amplification

After incubation, the cultures were centrifuged, to separate the supernatant. Total RNA was isolated from *A. fumigatus* mycelia by using the Direct-zol™ RNA MiniPrep kit (Zymo Research, Irvine, CA, USA); the manufacturer’s instructions were followed. cDNA was synthesized from previously extracted RNA; the reverse transcriptase enzyme (SuperScript^®^ II Reverse Transcriptase, Invitrogen) was used according to the manufacturer’s instructions.

The coding sequence of the Afu1g14710 gene (without the stop codon) was amplified from the synthesized cDNA through polymerase chain reactions (PCR) carried out with the enzyme Phusion High-Fidelity polymerase DNA Polymerase (Thermo Scientific, Waltham, MA, USA) in a thermocycler (Eppendorf). For this, cDNA was added to a mixture containing 1 U of Phusion HiFi enzyme, 1X HF buffer, 0.2 mM dNTP mix, 3% (*v*/*v*) DMSO, and 0.5 µM F and R primers. Specific primer sequences containing overlapping regions between the pPICZαA vector and the gene were applied (F: 5′–**GAGAAAAGAGAGGCTGAAGCTGAATTC**ATGGGCTCCACAACT–3′ and R: 5′–**ATCCTCTTCTGAGATGAGTTTTTGTTCTAG**AGCCTTCTCGATGTATTG–3′; overlapping sites in bold). The cycling conditions were 98 °C for 30 s; 30 cycles at 98 °C for 10 s, 55 °C for 30 s, and 72 °C for 1 min; 72 °C for 10 min and 4 °C–∞. The PCR product was analyzed by electrophoresis and purified from 1% (*w*/*v*) agarose gel by using the QIAquick Gel Extraction kit (Qiagen, Hilden, Germany).

### 3.4. Cloning, P. pastoris Transformation, and Screening for Recombinant Transformants

The fragment obtained by PCR was cloned into the pPICZαA vector (previously digested with restriction enzymes *EcoR*I and *Xba*I) by using the CPEC method (Circular polymerase extension cloning), according to the protocol described by [89]. For this purpose, the prepared fragment was added to a mixture containing the vector digested at 2:1 proportion, 1 U of Phusion HiFi enzyme, 1X HF buffer, 3% (*v*/*v*) DMSO, and 0.4 mM dNTPs mix. The cycling conditions were 98 °C for 30 s; 35 cycles of 98 °C for 10 s, 55 °C for 30 s, and 72 °C for 2 min; 72 °C for 10 min, and 4 °C–∞.

After that, the reaction product was transformed into competent *E. coli* DH10β (previously prepared), and the positive clones were confirmed by colony PCR. Next, the recombinant plasmid (Afu1g14710/pPICZαA) was linearized with *Pme*I and transformed into competent *P. pastoris* X-33 cells by electroporation, according to the EasySelect™ *Pichia* Expression Kit manual (Invitrogen). Positive clones were screened by Zeocin resistance.

### 3.5. Heterologous AfBgl1.3 Expression in P. pastoris and Protein Purification

One of the positive colonies was selected, pre-inoculated in 25 mL of BMGY broth (1% yeast extract, 2% peptone, 1% glycerol, 4 × 10^−5^% biotin, 1.34% nitrogen base for yeast, and 100 mM potassium phosphate buffer, pH 6.0), and incubated at 200 rpm and 30 °C for 16 h. After that, 2 mL of the pre-inoculum was transferred to 50 mL of BMGY liquid medium and incubated again at 200 rpm and 30 °C until O.D._600_ between 2 and 6 was reached. Then, the cells were transferred to a sterile 50-mL Falcon tube (enough volume to obtain O.D._600_ = 1 in 100 mL) and centrifuged at 3000× *g* and 25 °C for 5 min. Next, the supernatant was discarded, and the pellet was resuspended in 100 mL of BMMY broth (1% yeast extract, 2% peptone, 1.5% methanol, 1.34% nitrogen base for yeast, 4 × 10^−5^% biotin, and 100 mM potassium phosphate buffer, pH 6.0) to induce expression.

The culture was incubated at 220 rpm and 30 °C for seven days; 100% methanol was added to a final concentration of 1.5% to maintain the induction. After the induction time, the culture was centrifuged at 3000× *g* and 4 °C to for 5 min, to separate the supernatant. The supernatant was concentrated about 10-fold by using Amicon Ultra-15 with a cut-off of 10 kDa (Millipore, MA, USA).

For purification, the concentrate was transferred to 10 mL of 20 mM sodium phosphate binding buffer + 500 mM NaCl pH 7.4 and loaded into Ni Sepharose 6 Fast Flow resin (Ge Healthcare, Little Chalfont, United Kingdom). AfBgl1.3 was eluted with increasing concentrations of imidazole (0–500 mM) in binding buffer solution, which was used to wash the Ni Sepharose 6 Fast Flow resin. All the eluted fractions were collected and analyzed by 10% SDS-PAGE. Protein was quantified by the Greenberg method [90].

### 3.6. Analysis by Circular Dichroism (CD) and Intrinsic Tryptophan Fluorescence Emission (ITFE)

The AfBgl1.3 secondary structure was analyzed by CD on the JASCO-810 spectropolarimeter. After dilution in 20 mM sodium phosphate buffer (pH 7.4), 0.2 mL of 0.05 mg mL^−1^ AfBgl1.3 was incubated at different temperatures (25–80 °C) for 30 min. The samples were placed in a quartz cuvette with optical path length of 1 mm and read in quadruplicate from 190 to 250 nm (far UV). Average spectra, from which the spectrum of the protein-free buffer (blank) was subtracted, were obtained. The parameters were scan rate of 50 nm min^−1^, bandwidth of 3 nm, and D.I.T of 1s. The resulting spectra were converted from millidegrees (mdeg) to Δε (M^−1^ cm^−1^) according to the following equation: Δε = θ[(0.1⋅MRW)/(d⋅c⋅3298)], where θ is the ellipticity value given by the equipment; MRW is the mean residual weight of AfBgl1.3; c is the enzyme concentration (mg mL^−1^); and d is the optical path length (cm).

Secondary structure percentages were determined using the free program BeStSel [31] and the results were compared with the structure modeled in the server Phyre2 [33] and program Biovia Discovery Studio Visualizer version 20.1.0 (Dassault Systèmes, San Diego, CA, USA).

ITFE was analyzed on the HITACHI-F4500 spectrofluorimeter. Tryptophan fluorescence was measured by placing 1 mL of the same samples in quartz cuvettes with optical path length of 1 cm. The samples were excited at 295 nm, and fluorescence emission was detected from 300 to 450 nm. Readings were taken at 240 nm min^−1^, every 0.2 nm. The average spectra (obtained in quadruplicate) were also corrected by subtracting the spectrum of the blank.

To determine the apparent T_m_, the first derivative of the ellipticity curve at 230 nm versus temperature (in the case of circular dichroism) and the first derivative of the maximum wavelength versus temperature (in the case of intrinsic tryptophan emission) were obtained by using the program Origin 9.0. All the CD and ITFE plots were previously smoothed with the Savitzky-Golav filter [91] and fitted with the Boltzmann function.

### 3.7. AfBgl1.3 Activity Assay

The activity of purified AfBgl1.3 was determined by different methods depending on the substrate. All the reactions were performed in a final volume of 100 µL, in 50 mM sodium phosphate buffer (pH 6.0) containing 0.1 µg of the enzyme and 1% (*w*/*v*) of each substrate at 40 °C for 30 min.

The AfBgl1.3 activity toward the non-reducing substrate, D-(-)-salicin (Sigma-Aldrich), was calculated by quantifying the glucose released during the reaction by using the DNS method [92].

By employing the colorimetric substrate *p*-nitrophenyl-β-D-glucopyranoside (Sigma-Aldrich), the AfBgl1.3 activity was evaluated by quantifying the *p*-nitrophenol product released during the reaction. To this end, a calibration curve was constructed for *p*-nitrophenol under the same reaction conditions. The reaction was stopped by adding 2 M Na_2_CO_3_ to the reaction medium. Absorbance was read on a spectrophotometer at λ = 405 nm.

When the reducing substrates cellobiose (Sigma-Aldrich) and lactose (Sigma-Aldrich) were used, the AfBgl1.3 activities were determined by quantifying the glucose released during the reaction by means of the GOD-PAP Liquid Stable Glucose Kit (LABORLAB, Lafayette, CO, USA); the manufacturer’s recommendations were followed.

One unit (U) of AfBgl1.3 was considered as the amount of enzyme required for hydrolyzing 1 µmol of each substrate per minute under each of the conditions described above.

### 3.8. AfBgl1.3 pH and Temperature Profiles

To characterize the AfBgl1.3 pH and temperature profiles, the substrate D-(-)-salicin was used according to the protocol described above.

The optimal pH for AfBgl1.3 activity was determined for pH ranging from 3.0 to 8.0. For this purpose, the reaction medium consisted of McIlvaine buffer (citric acid-Na_2_HPO_4_) at the different evaluated pH. The reaction was conducted at 40 °C for 30 min.

The optimal temperature for AfBgl1.3 activity was determined between 20 and 60 °C. To this end, reactions were performed at different temperatures for 30 min.

AfBgl1.3 pH stability was estimated by measuring the residual enzymatic activity of purified AfBgl1.3 under optimal conditions. First, in the absence of substrate, AfBgl1.3 was pre-incubated at pH ranging from 3.0 to 8.0 in McIlvaine buffer for 24 or 48 h. AfBgl1.3 activity without pre-incubation was considered 100%.

AfBgl1.3 thermal stability was estimated by measuring the residual enzymatic activity at pH 6.0 and 40 °C after AfBgl1.3 had been pre-incubated without substrate at 30, 40, 45, or 50 °C for 1, 3, 6, 15, or 24 h. AfBgl1.3 activity without pre-incubation was considered 100%.

To assess AfBgl1.3 stability at refrigerator temperature (4 °C), AfBgl1.3 was stored in 20 mM sodium phosphate binding buffer + 500 mM NaCl (pH 7.4) for up to four months. Then, the relative activity was determined. The AfBgl1.3 activity on the first day after purification was 100%.

### 3.9. Influence of Additives on AfBgl1.3 Activity

To determine how metal ions and chemical reagents (Mg^2+^, Zn^2+^, Mn^2+^, Ni^2+^, Co^2+^, Cu^2+^, Ag^+^, Ca^2+^, Fe^2+^, Na^+^, K^+^, NH_4_^+^, β-mercaptoethanol, EDTA, SDS, DTT, DMSO, and Tween 20) affected the AfBgl1.3 activity, AfBgl1.3 was incubated in the reaction mixture containing one of the additives at 5 mM. The reactions were carried out at 40 °C for 30 min. AfBgl1.3 activity without additive was considered 100%.

### 3.10. Effects of Glucose on AfBgl1.3 Activity

The effect of product inhibition on AfBgl1.3 activity was determined in the presence of increasing glucose concentrations (Sigma-Aldrich) (0–3000 mM). For this, the reaction medium consisted of 0.1 µg of AfBgl1.3, 1% (*w*/*v*) chromogenic substrate *p*-nitrophenyl-β-D-glucopyranoside (Sigma-Aldrich), 50 mM sodium phosphate buffer (pH 6.0), and different glucose concentrations. The reactions were accomplished at 40 °C for 30 min. AfBgl1.3 activity in the absence of glucose was considered 100%.

### 3.11. Kinetic Assays

The AfBgl1.3 K_M_ and V_max_ values were determined by using increasing concentrations of the substrates D-(-)-salicin, *p*-nitrophenyl-β-D-glucopyranoside and cellobiose (0–30 mg mL^−1^). The reactions were carried out as described in Section 3.7, and AfBgl1.3 activity was measured according to the method proposed for each substrate. The kinetic parameters were determined by non-linear regression; the program Origin 9.0 was employed.

### 3.12. Determination of AfBgl1.3 Transglycosylation Activity by Thin Layer Chromatography (TLC)

The AfBgl1.3 transglycosylation activity was evaluated by adding 0.1 µg of AfBgl1.3 to reaction medium containing 1% (*w*/*v*) cellobiose, 50 mM sodium phosphate buffer (pH 6), and 0, 10, 20, or 45 g L^−1^ glucose. Reactions were incubated at 40 °C for 30 or 60 min and stopped by incubation at 98 °C for 5 min. The same experiment was carried out in the absence of cellobiose for 30 min.

Chromatography was accomplished on a Kiesegel 60 (Merck) silica gel plate (20 cm × 20 cm) according to Fontana et al., 1988 [93]. After the reactions, 5 µL of each sample was applied to the plate. To separate the reaction components, the plate was subjected to two chromatographic runs; ethyl acetate/acetic acid/formic acid/water (9:3:1:4, *v*/*v*/*v*/*v*) was employed as mobile phase. After each run, the plate was dried at room temperature for 24 h; after the second run, the plate was developed by adding the developer solution consisting of 0.4% orcinol (Sigma-Aldrich) in an ethanol mixture and concentrated H_2_SO_4_ (9:1, *v*/*v*). After the plate was dried at room temperature, it was incubated in an oven at 70 °C until reddish stains, corresponding to the different products, appeared.

### 3.13. Determination of AfBgl1.3 Transglycosylation Products by High Performance Liquid Chromatography (HPLC)

Cellobiose hydrolysis was also analyzed by ion chromatography (HPLC-Thermo Fisher-U3000), coupled with refractive index detector and Rezex-ROA column (Phenomenex) (eluted with 5 mM H_2_SO_4_, at flow rate 0.6 mL/min and column temperature 45 °C). For this, 1 µg of AfBgl1.3 was incubated with 1% (*w*/*v*) cellobiose and 50 mM sodium phosphate buffer (pH 6), for 30 min, in a final volume of 1 mL. The reaction was filtered, dried and resuspended in the mobile phase. Cellotriose, cellobiose and glucose was identified by comparing its retention time with those of an authentic standard [94].

### 3.14. Activity with Celluclast^®^ 1.5L by AfBgl1.3 Addition

Enzymatic hydrolysis of carboxymethyl cellulose (CMC) (Sigma-Aldrich) was carried out in reaction medium containing 50 mM sodium phosphate buffer (pH 6.0), 1% (*w*/*v*) substrate, and 2.5 μg of AfBgl1.3/g of CMC.

Hydrolysis was performed by supplementing the commercial cocktail Celluclast^®^ 1.5L (Sigma-Aldrich) at two different concentrations: 0.1 and 0.9 FPU/g of substrate. The reaction was accomplished in the ThermoMixer (Eppendorf) in a final volume of 1 mL at 1000 rpm and 40 °C for 6 or 12 h. The experiment control was prepared under the same conditions, but without AfBgl1.3. Given the reaction time, the concentration of reducing sugars released in the reaction was determined by the DNS method, at 540 nm. All the experiments were replicated at least four times.

### 3.15. Combined Assays

The effect of the synergistic activity of AfBgl1.3 and the cellulases endoglucanase (Af-EGL7) and cellobiohydrolase (AfCel6A) [82,83] was evaluated during sugarcane delignified bagasse (SBD) and CMC hydrolysis. The reactions were conducted in 50 mM sodium phosphate buffer (pH 6.0) containing 1% (*w*/*v*) of each substrate at 1000 rpm and 40 °C for 24 h.

CMC hydrolysis was performed by using different proportions of the enzymes Af-EGL7, AfCel6A, and AfBgl1.3 (1:10:1, 1:10:10, 1:10:25, or 1:10:50); the amount of AfBgl1.3 added per gram of cellulose varied (1, 10, 25, or 50 µg), while the amounts of Af-EGL7 and AfCel6A were fixed at 1 and 10 µg per gram of substrate. For SDB hydrolysis, the amounts of the enzymes were Af-EGL7 = 4 µg, AfCel6A = 40 µg, and AfBgl1.3 = 200 µg. The percent hydrolysis yields were determined by estimating the released reducing sugars by the DNS method. The reported results represent the means ± SD calculated from at least three experimental replicates.

## 4. Conclusions

We have described the isolation, characterization, and heterologous expression of a novel GH1 β-glucosidase (AfBgl1.3) from the thermophilic fungus *A. fumigatus*. The tolerance of the AfBgl1.3 to glucose is interesting for industrial applications because glucose-tolerant β-glycosidases are not inhibited by feedback. Besides that, the experimental results show that AfBgl1.3 has transglycosylation activity, forming cellotriose in the presence of cellobiose. When we supplemented an enzyme cocktail for CMC and SDB saccharification with AfBgl1.3, the glucose-tolerant enzyme improved the hydrolysis efficiency. Therefore, AfBgl1.3 has potential application value.

## Figures and Tables

**Figure 1 ijms-24-04489-f001:**
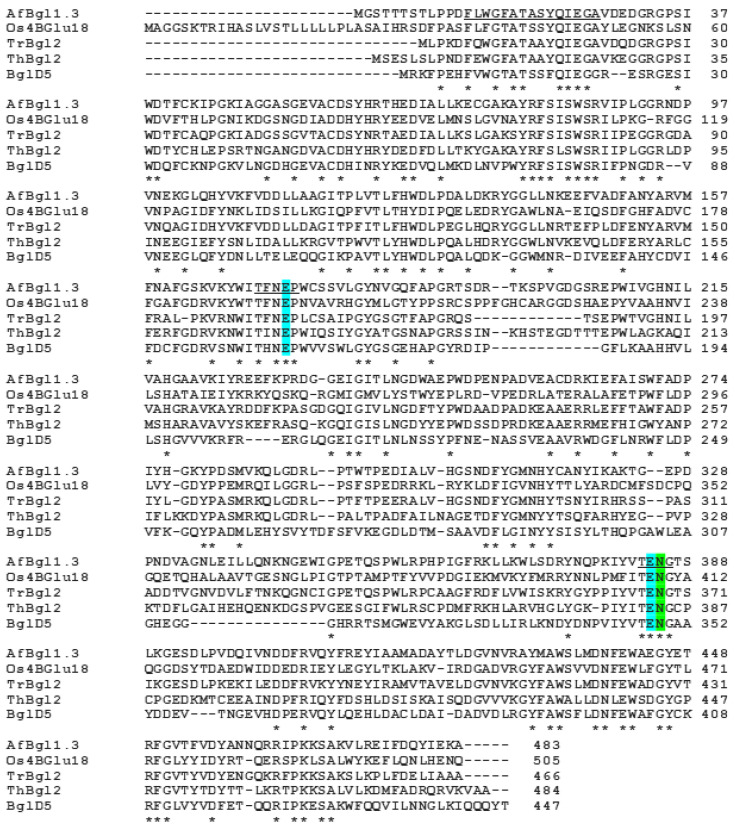
Multiple sequence alignment of AfBgl1.3 with other GH1 β-glycosidases, such as *Oryza sativa* Os4BGlu18 (Q7XSK0), *Trichoderma reesei* TrBgl2 (O93785), *Trichoderma harzianum* ThBgl2 (A0A0F9XM91), and *Jeotgalibacillus malaysiensis* BglD5 (A0A0B5ARU7) was performed using the tool Clustal Omega l. An asterisk indicates the positions with single, fully conserved sequences. Blue: Potential catalytic sites, Green: *N*-glycosylation site (_): motifs.

**Figure 2 ijms-24-04489-f002:**
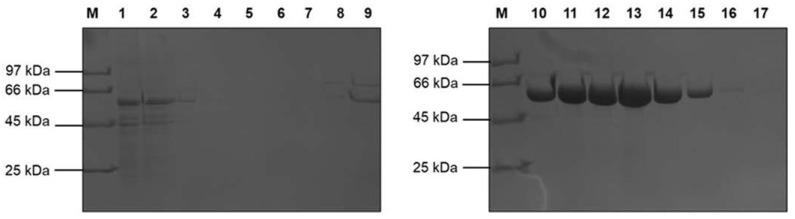
SDS-PAGE 10% analysis after recombinant AfBgl1.3 purification. M: Molecular mass marker (Low Molecular Weight Electrophoresis Calibration Kit—GE Healthcare); Lanes 1: Flow; 2–6: Washing with 20 mM sulfur phosphate solution containing 500 mM NaCl (pH 7.4); 7, 8, and 9: AfBgl1.3 elution with 5, 10, and 20 mM imidazole, respectively; 10 and 11: recombinant AfBgl1.3 elution with 40 mM imidazole; 12 and 13: recombinant AfBgl1.3 elution with 80 mM imidazole; 14 and 15: recombinant AfBgl1.3 elution with 160 and 250 mM imidazole, respectively; 16 and 17: recombinant AfBgl1.3 elution with 500 mM imidazole. Proteins were developed with Coomassie Brilliant Blue G-250.

**Figure 3 ijms-24-04489-f003:**
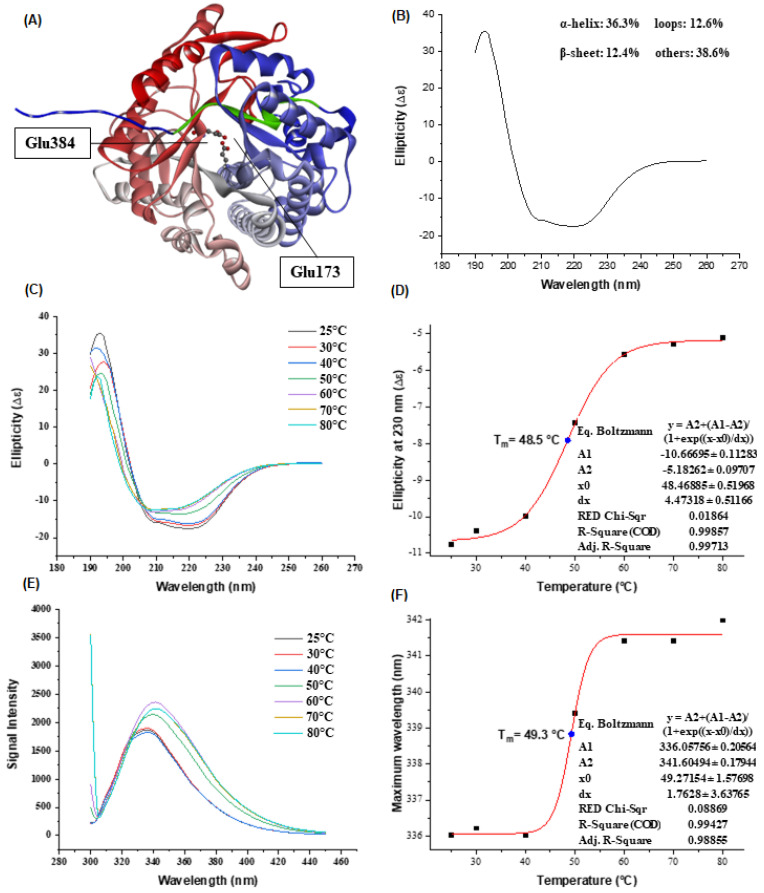
AfBgl1.3 structural analysis. (**A**) AfBgl1.3 predicted tridimensional structure. The N-terminal motif FLWGFATASYQIEGA is highlighted in green. The residues Glu384 and Glu173 are indicated with black lines and are depicted in balls and sticks. The tertiary structure is colored from the N-terminal (blue) to the C-terminal (red) domains. (**B**) Far-UV CD spectrum at 25 °C. (**C**) Far-UV CD spectrum after AfBgl1.3 incubation at 25–80 °C for 30 min. (**D**) First derivative of the ellipticity at 230 nm by temperature, to determine the apparent Tm. (**E**) ITFE spectrum after AfBgl1.3 incubation at different temperatures from 25 to 80 °C for 30 min. (**F**) First derivative of the maximum wavelength by temperature, to determine the apparent Tm.

**Figure 4 ijms-24-04489-f004:**
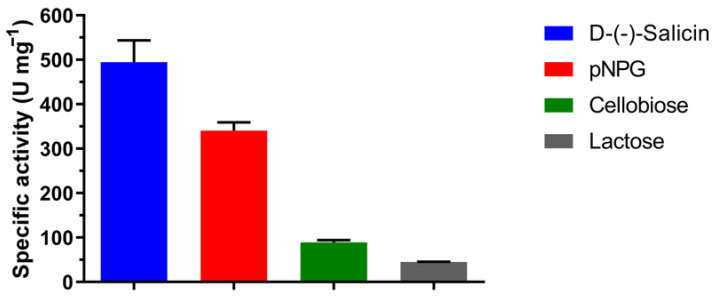
AfBgl1.3 specific activity (U mg^−1^) toward D-(-)-salicin, pNPG, cellobiose, and lactose. Each reaction was performed in 50 mM phosphate buffer (pH 6.0) containing 1.0% (*w*/*v*) of each substrate at 40 °C for 30 min. Values are the mean ± SD of three replicates.

**Figure 5 ijms-24-04489-f005:**
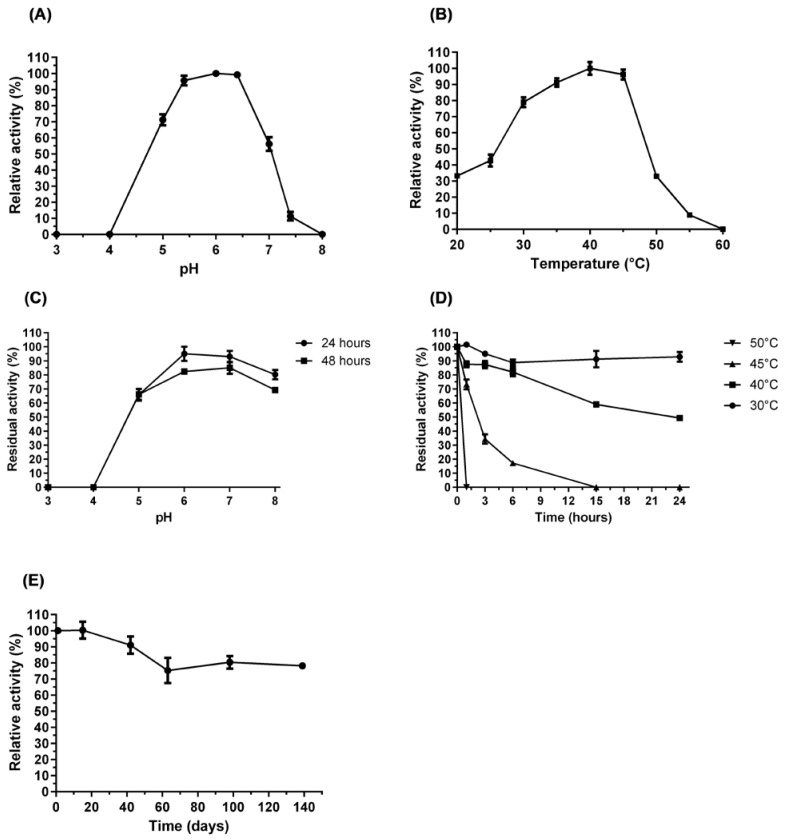
Stability and optimal conditions for recombinant AfBgl1.3 activity. (**A**) pH curve. (**B**) Temperature curve. (**C**) pH stability. (**D**) Thermostability. (**E**) Stability in the refrigerator (4 °C).

**Figure 6 ijms-24-04489-f006:**
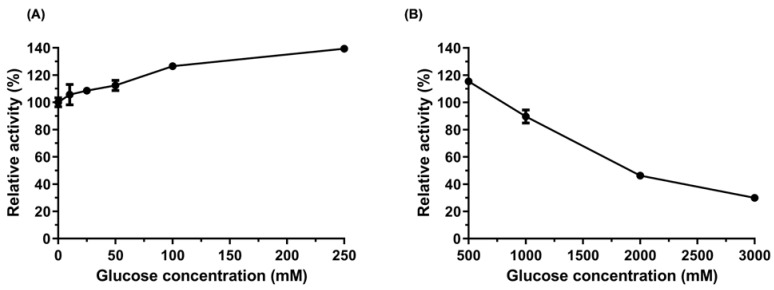
Effect of D-glucose on the AfBgl1.3 activity toward pNPG. (**A**) 0–250 mM glucose. (**B**) 500–3000 mM glucose. Error bars, S.D.

**Figure 7 ijms-24-04489-f007:**
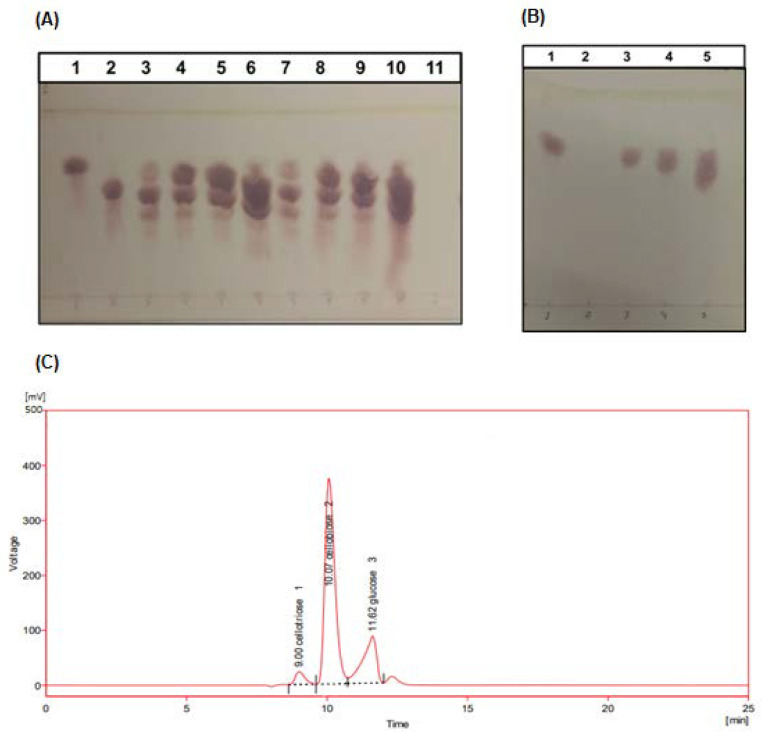
Verifying the presence or absence of a transglycosylation product. The reaction products were analyzed by TLC on activated silica plates by using an ethyl acetate/acetic acid/formic acid/water eluent (9:3:1:4) and HPLC, coupled with refractive index detector and Rezex-ROA column (Phenomenex) (eluted with 5 mM H_2_SO_4_, at flow rate 0.6 mL/min and column temperature 45 °C). (**A**) Enzymatic product analysis of substrates 1% cellobiose and 0, 10, 20, or 45 g/L glucose measured after 30 and 60 min. *Lane* 1: glucose standard; *Lane* 2: cellobiose standard; *Lanes* 3, 4, 5, and 6: products from the reaction in the presence of 0, 10, 20, and 45 g/L glucose, respectively, after 30 min; *Lanes* 7, 8, 9, and 10: products from the reaction in the presence of 0, 10, 20, and 45 g/L glucose, respectively, after 60 min; *Lane* 11: AfBgl.3 in 50 mM sodium phosphate buffer pH 6.0. (**B**) Similar assays were carried out as in A, but just in the presence of glucose. *Lane* 1: glucose standard; *Lane* 2: AfBgl.3 in 50 mM sodium phosphate buffer pH 6.0; *Lanes* 3, 4, and 5: products from the reaction in the presence of 10, 20, and 45 g/L glucose, respectively, after 30 min. No transoligosaccharide product band was observed in any of the reactions, indicating that transglycosylation does not occur when only exogenous glucose is present. (**C**) Analysis of cellobiose hydrolysis products by HPLC.

**Figure 8 ijms-24-04489-f008:**
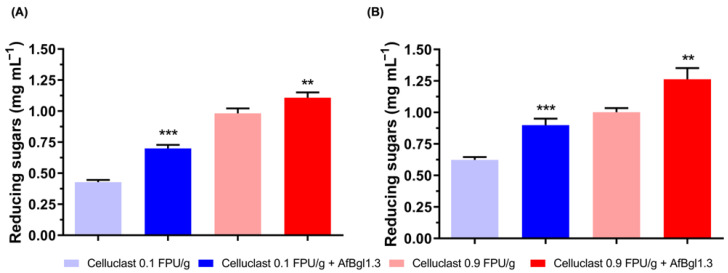
Evaluation of the addition of AfBgl1.3 in the commercial cocktail Celluclast^®^ 1.5L during CMC degradation. (**A**) Hydrolysis for 6 h. (**B**) Hydrolysis for 12 h. Values are the mean ± SD of four experimental replicates. Asterisks indicate significant difference in relation to the control system (cocktail alone), *p* ≤ 0.01 (**) and *p* ≤ 0.001 (***).

**Figure 9 ijms-24-04489-f009:**
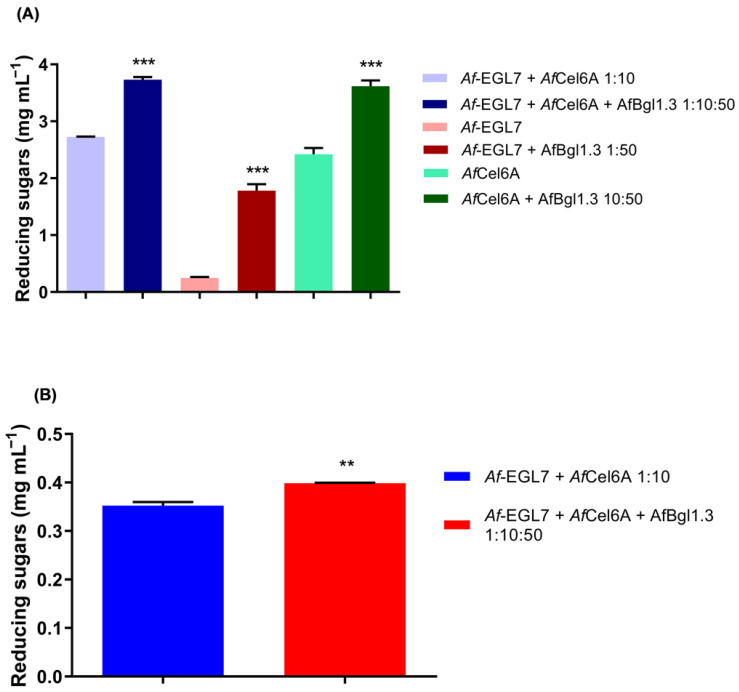
Evaluation of synergism for AfBgl1.3, Af-EGL7, and AfCel6A during (**A**) CMC and (**B**) SBD hydrolysis. Values are given as the mean ± standard deviation of four experimental replicates. Asterisks indicate significant differences between enzyme associations at different proportions with respect to controls (Af-EGL7 and AfCel6A), *p* ≤ 0.01 (**) and *p* ≤ 0.001 (***).

**Table 1 ijms-24-04489-t001:** Effect of temperature on the proportion of secondary structures in AfBgl1.3.

Temperature (°C)	α-Helices (%)	β-Sheets Antiparallel (%)	β-Sheets Parallel (%)	Loops (%)	Others and Disorderly (%)
25	36.62	3.97	13.84	11.04	34.53
30	38.9	0.43	13.17	11.83	35.67
40	34.72	5.99	13.23	11.84	34.22
50	31.27	9.28	12.57	11.31	35.56
60	27.23	17.43	7.54	10.14	37.66
70	21.16	19.09	10.17	11.14	38.44
80	22.83	16.75	7.53	11.47	41.37

**Table 2 ijms-24-04489-t002:** Comparison of biochemical properties and kinetics of GH1 β-glucosidases from different fungi.

*Dictyoglomus turgidum*	*T. halotolerans* YIM 90462T	*Trichoderma reesei*	*Anoxybacillus thermarum*	*Saccharomonospora* sp. NB11	*Microbulbifer* sp. ALW1	*Bacillus* sp. CGMCC 1.16541	*Jeotgalibacillus malaysiensis*	*Paecilomyces thermophila*	*Aspergillus fumigatus* Af293	Source Organism
DturβGlu	ThBGL1A	bgl1	BgAt	BglNB11	MaGlu1A	BsBgl1A	BglD5	PtBglu1	AfBgl1.3	Protein
*E. coli* BL21(DE3)-RIL	*E. coli* BL21	*Pichia pastoris* GS115	*E. coli* BL21 (DE3)	*E. coli* BL21 (DE3)	*E. coli* BL21 (DE3)	*E. coli* BL21 (DE3)	*E. coli* BL21 (DE3)	*Pichia pastoris* GS115	*Pichia pastoris* X-33	**Expression system**
80 °C	40–55 °C	70 °C	65 °C	40 °C	40 °C	45 °C	65 °C	55 °C	35–45 °C	**Optimum Temperature**
5.4	5.6–6.6	5	7	7	4.5	5.6–7.6	7	6	5.4–6.4	**Optimum pH**
After 2 h of pre-incubation at 70 °C and 80 °C, the residual activity was 70% and 50%, respectively	100% of its residual activity at 40 °C after 2 h;	Residual activity >90% at 60 °C for 60 min	Residual activity about 100% after 24 h at 50 °C;	40% of residual activity at 40 °C for 2 h and 70% of residual activity after 2 h in the temperature range of 25−35 °C	34.0% of residual activity after 0.5 h at 40 °C	Residual activity about 100% after 120 min at 45 °C	t1/2 65 °C = 35 min (with calcium); t1/2 65 °C = 70 min (without calcium)	Residual activity about 88% activity after 30 min at 55 °C	Residual activity about 60% after 15 h at 40 °C and 90% residual activity after 24 h at 30 °C	**Thermostability**
Residual activity remained close to 90% after 1 h of pre-incubation in the pH range 5.0–8.0	Residual activity >60% after 12 h and 24 h pre-incubation at pH 5–8	Residual activity >90% in the pH range 4–7	Enzyme activity induction in the pH range 4.0–6.5 after 15min of pre-incubation	Residual activity >50% after 8 h pre-incubation in the pH range 6–10	-	Residual activity about 100% in the pH range (4–9) after 24 and 48 h of pre-incubation	-	Residual activity >80% in the pH range (5.0–11.0)	Residual activity >70% after 24 and 48 h of incubation in the pH range 5–8	**pH stability**
pNPG Salicin	pNPG Cellobiose	-	pNPG	pNPG	pNPG Cellobiose	pNPG Cellobiose	pNPG	pNPG Salicin Cellobiose	pNPG Salicin Cellobiose	**Substrate**
-	52.6 U mg^−1^ 33.8 U mg^−1^	-	7614 U mg^−1^	5735.8 U mg^−1^	4.52 U mg^−1^ 151.52 U mg^−1^	36 ± 0.6 U mg^−1^ 78 ± 2 U mg^−1^	39.48 ± 0.63 U mg^−1^	328.8 ± 7.5 U mg^−1^ 66.0 ± 0.4 U mg^−1^ 306.3 ± 6.3 U mg^−1^	656.0 ± 17.5 U mgˉ^1^ 706.5 ± 23.8 U mgˉ^1^ 132.6 ± 7.1 U mgˉ^1^	**V_max_**
0.84 mM 8.12 mM	21.96 mM 3.06 mM	-	0.360 mM	0.4037 mM	2.71 mM 24.44 mM	9 ± 0.2 mM 0.11 ± 0.02 mM	0.50 ± 0.02	0.55 ± 0.03 mM 6.85 ± 0.06 mM 1.0 ± 0.06 mM	7.6 ± 0.8 mM 17.6 ± 1.9 mM 15.4 ± 2.5 mM	**K_M_**
8710 s^−1^ 659 s^−1^	41.8 s^−1^ 30 s^−1^	-	0.63 × 10^4^ s^−1^	5042.16 s^−1^	-	31.3 ± 0.5 s^−1^ 67.8 ± 1.7 s^−1^	33.93 ± 0.54 s^−1^	5.5 s^−1^ 1.1 s^−1^ 5.1 s^−1^	595.8 s^−1^ 621.7 s^−1^ 120.4 s^−1^	** *kcat* **
1 × 10^4^ mM^−1^ s ^−1^ 81 mM^−1^ s ^−1^	1.9 s^−1^ mM^−1^ 9.8 s^−1^ mM^−1^	-	1.74 × 10^4^ s^−1^ mM^−1^	12,487.71 s^−1^ mM^−1^	-	3.5 ± 0.1 s^−1^/mg/mL 616 ± 2 s^−1^/mg/mL	-	9.96 mM^−1^ s^−1^ 0.16 mM^−1^ s^−1^ 5.10 mM^−1^ s^−1^	7.8 × 10^4^ Mˉ^1^ sˉ^1^ 3.5 × 10^4^ Mˉ^1^ sˉ^1^ 7.8 × 10^3^ Mˉ^1^ sˉ^1^	** *kcat* ** **/K_M_**
[48]	[49]	[50]	[51]	[52]	[53]	[54]	[28]	[55]	This work	**Reference**

**Table 3 ijms-24-04489-t003:** Kinetic constants of recombinant AfBgl1.3.

Substrate	*V*_max_ (U mg^−1^)	K_M_ (mM)	*k_cat_* (s^−1^)	*k_cat_*/K_M_ (M^−1^s^−1^)
pNPG	656.0 ± 17.5	7.6 ± 0.8	595.8	7.8 × 10^4^
Salicin	706.5 ± 23.8	17.6 ± 1.9	621.7	3.5 × 10^4^
Cellobiose	132.6 ± 7.1	15.4 ± 2.5	120.4	7.8 × 10^3^

**Table 4 ijms-24-04489-t004:** Effect of different ions and reagents at 5 mM on recombinant AfBgl1.3 activity.

Additive	Relative Activity (%)	Additive	Relative Activity (%)
Control	100.0 ± 3.0	ZnSO_4_	0
(NH_4_)_2_SO_4_	108.7 ± 1.2	EDTA	94.5 ± 2.3
MnCl_2_	114.5 ± 1.2	β-mercaptoethanol	103.5 ± 3.8
KCl	108.4 ± 2.1	DMSO	116.3 ± 3.9
MgSO_4_	91.4 ± 0.0	DTT	111.2 ± 2.3
CaCl_2_	83.8 ± 1.7	Triton X-100	123.1 ± 2.6
CoCl_2_	39.9 ± 5.4	Tween 20	118.2 ± 2.5
FeSO_4_	31.4 ± 1.1	SDS	0
AgNO_3_	0	SLS	0
CuSO_4_	0	Ascorbic acid	0

## Data Availability

Data sharing is not applicable to this article.

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
