# Peer review of "Characterization of a New Glucose-Tolerant GH1 β-Glycosidase from *Aspergillus fumigatus* with Transglycosylation Activity"

_ijms, 2023, doi:10.3390/ijms24054489_

Round 1
Reviewer 1 Report
Reviewer’s report on “Characterization of a new glucose-tolerant β-glycosidase GH1 from Aspergillus fumigatus with transglycosylation activity”
General comments:
Advances from this work have implications for improving the valorization agricultural residues to value-added products, driving the bioeconomy forward. As a result, of the significance, well conceptualized and executed study, I recommend acceptance of the manuscript for publication upon satisfactorily addressing of the listed major comments. However, numerous technical flaws, which may require further experimental work have been noted in the manuscript. Listed below are the technical and conceptual shortcomings of the work which should be addressed by the authors.
Conceptual and technical comments:
The authors are missing a sentence touching on the significance of embarking on this study, I suggest that this be added as the preamble problem statement as the first sentence in the abstract.
It would seem that the transglycosylation activity of AfBgl1.3 is counterproductive for applications in biomass saccharification, as this would lead to the reduction of the fermentable monomeric sugar, glucose. Can the authors justify the use of such an enzyme in the saccharification of biomass?
In section 2.2, lines 125-126: the authors say “Treatment with Endo H (New England Biolabs, Ipswich, MA, USA) did not reduce the AfBgl1.3 molecular mass because one site is not enough to reduce a molecular mass that can be seen on SDS-PAGE”. Endo H cleaves asparagine-linked mannose-rich oligosaccharides, but not highly processed complex oligosaccharides from glycosylated proteins. Is it possible that complex glycosylation could have occurred in this protein?
In this study, an in-silico method was used to predict the structure of the protein of AfBgl1.3. However, the validation and analysis, and subsequent refinement of this predicted structure were lacking. For example, the stereochemical quality of protein structures using all-atom contact analysis tools and updated geometrical criteria for phi/psi, sidechain rotamer, and Cbeta deviation can be achieved by using Procheck or Verify3D.
With respect to the effect of chemical reagents and metal ions on the enzymes, statistical analysis (t-test or ANOVA) needs to be included, otherwise, authors cannot claim whether there is any significant difference between samples.
In lines 322-323, the authors say “Except glucose, we detected no sugar, which indicated that transglycosylation did not occur when only glucose was present”, yet lanes 3-5 show spots with variable Rf values on the TLC plate that are different to lane 1 (glucose alone). Can the authors revise the result narrative and discussion in this regard?
In lanes 335-336, the authors say “when the enzyme is incubated with cellobiose only, the main product of the reaction is glucose, and no transglycosylation product is detected.”, yet, lanes 3 and 7 in Fig7A clearly shows that cellobiose produced both glucose and cellotriose in the presence of the enzyme. Can the authors revise the result narrative and discussion in this regard?
The authors tested the commercial cellulase cocktail, Celluclast 1.5L, known to be concentrated with higher cellobiohydrolase activity, particularly Cel7A, on CMC, a substrate meant to test endo-glucanase activity. This is not the best choice of the substrate to test for the complete cocktail’s suite of activities, substrates such as Avicel PH101 or Bacterial crystalline cellulose would have sufficed for this. It is also known that native cellulose is rich in crystalline regions than amorphous regions, therefore, results from CMC cannot be translated into a real-world context. In this regard, the experimental setup for the enzyme synergy studies is flawed.
Also, Celluclast 1.5L contains minimal glucosidase activity, hence the suppliers recommend using it with Novozyme 188, a glucosidase preparation, to achieve the conversion of cellooligosaccharides to glucose. Therefore, the addition of the glycosidase in this study may not necessarily be synergistic but merely additive. To confirm this, the authors should monitor glucose release by Celluclast 1.5L in the presence and absence of glycosidase.
Grammar and specific comments:
Title:
The title should be revised to “Characterization of a new glucose-tolerant GH1 β-glycosidase from Aspergillus fumigatus with transglycosylation activity”
Abstract:
Line 19, change “stimulation” to “co-incubation”
Line 27, change “in synergy” to “synergistically”
Introduction:
Line 46-49, rewrite as “reducing end-acting cellobiohydrolases (EC 3.2.1.176) and non-reducing end-acting cellobiohydrolases (EC 3.2.1.91) release cellooligosaccharide units like cellobiose”
Results and discussion:
Line 100, rewrite as “Multiple sequence alignment”
Line 107-109, describe “X” as a variable amino acid in the sequence motifs for the reader
Line 117, rewrite as “Expression of recombinant β-glycosidase, AfBgl1.3, in P. pastoris and its characterization”
Line 124, rewrite as “GH1 β-glycosidases”
Line 147, change “submitting” to “subjecting”
Line 153-154, the GH family affiliation comes before the enzyme’s name
Line 220, change “including” to “such as”
Line 365, fix “released sugars release”
Line 629, rewrite as “for CMC and SDB saccharification”
Author Response
Advances from this work have implications for improving the valorization agricultural residues to value-added products, driving the bioeconomy forward. As a result, of the significance, well conceptualized and executed study, I recommend acceptance of the manuscript for publication upon satisfactorily addressing of the listed major comments. However, numerous technical flaws, which may require further experimental work have been noted in the manuscript. Listed below are the technical and conceptual shortcomings of the work which should be addressed by the authors.
Conceptual and technical comments:
The authors are missing a sentence touching on the significance of embarking on this study, I suggest that this be added as the preamble problem statement as the first sentence in the abstract.
Reply: We are truly grateful for your critical comments. We´ve added the sentence “Concern over environmental impacts has spurred many efforts to replace fossil fuels with biofuels such as ethanol. However, for this to be possible, it is necessary to invest in other production technologies, such as second generation (2G) ethanol, in order to raise the levels of this product and meet the growing demand. Currently, this type of production is not yet economically feasible, due to the high costs of the enzyme cocktails used in saccharification stage of lignocellulosic biomass. In order to optimize these cocktails, the search for enzymes with superior activities has been the goal of several research groups. For this end, we have characterized the new β-glycosidase AfBgl1.3 from A. fumigatus after expression and purification in Pichia pastoris X-33”
It would seem that the transglycosylation activity of AfBgl1.3 is counterproductive for applications in biomass saccharification, as this would lead to the reduction of the fermentable monomeric sugar, glucose. Can the authors justify the use of such an enzyme in the saccharification of biomass?
Reply: We are truly grateful for your critical comments and thoughtful suggestions. In this article we are characterizing for the first time AfBgl1.3 and thus, we show all the possible applications and activities of the enzyme. In fact, it could be counterproductive its application in biomass saccharification, but this was not observed. We verified a considerable increase in the release of reducing sugars when AfBgl1.3 was associated with the Celluclast® 1.5L and also with the enzymes Af-EGL7, AfCel6A (Fig 8 and Fig. 9). Furthermore, the amount of cellotriose formed is very low (Fig. 7C) when compared to the release of reducing sugars, showing again that both activities can occur in the presence of cellobiose. These results show an advantage in this profile of AfBgl1.3, because transglycosylation of cellobiose may prevent its accumulation, avoiding inhibition of other enzymes and resulting in slightly higher glucose yields. Several papers have reported that Glucose-stimulated beta-glucosidases that do not present transglycosylation activity (Liu et al., 2017; Uchiyama et al., 2015) are inhibited by low glucose concentrations.
In section 2.2, lines 125-126: the authors say “Treatment with Endo H (New England Biolabs, Ipswich, MA, USA) did not reduce the AfBgl1.3 molecular mass because one site is not enough to reduce a molecular mass that can be seen on SDS-PAGE”. Endo H cleaves asparagine-linked mannose-rich oligosaccharides, but not highly processed complex oligosaccharides from glycosylated proteins. Is it possible that complex glycosylation could have occurred in this protein?
Reply: We didn´t identify any other glycosylation sites in the protein besides asparagine at position 385. Thus, we believe that it is the only potentially glycosylated amino acid, since the size difference between the predicted molecular weight (54KDa) and that observed on SDS-PAGE was not high.
In this study, an in-silico method was used to predict the structure of the protein of AfBgl1.3. However, the validation and analysis, and subsequent refinement of this predicted structure were lacking. For example, the stereochemical quality of protein structures using all-atom contact analysis tools and updated geometrical criteria for phi/psi, sidechain rotamer, and Cbeta deviation can be achieved by using Procheck or Verify3D.
Reply: We are truly grateful for your critical comments. We’ve improved the analysis and validation, and added the sentence “AfBgl1.3 tridimensional structure was predicted using AlphaFold [84] through the cloud computing platform ColabFold v1.5.1 [85]. The amino acid FASTA sequence for the corresponding protein was retrieved from the NCBI database (https://www.ncbi.nlm.nih.gov/protein/), under the accession number XP_752840.1. and was submitted to ColabFold for model prediction with the application of AMBER force field for the relaxation of the firstly ranked modeled structure. The structure was built based on MMseqs2 homology search with 20 recycles. The best-ranked model was selected and submitted to the web server GalaxyRefine (https://galaxy.seoklab.org/cgi-bin/submit.cgi?type=REFINE) [86], for structure refinement with default parameters and the retrieved model was submitted for validation through PROCHECK [87,88]. The model was properly validated (see Supplementary file S1) and the structure was visualized and analyzed on BIOVIA Discovery Studio Visualizer v21.1.0.20298” in item 3.2
With respect to the effect of chemical reagents and metal ions on the enzymes, statistical analysis (t-test or ANOVA) needs to be included, otherwise, authors cannot claim whether there is any significant difference between samples.
Reply: Thank you very much for your suggestion. However, the values presented are calculated from the relative activity, which shows how much the enzyme activities increased in percentage in the presence of additives, which does not make statistical calculation possible. For this calculation, the absolute data of enzyme activity should be presented. The presentation of results like this, without statistical analysis, is well accepted in the literature, according to articles published by our research group: Bernardi, A.V., Ind Crops Prod 2021, 170, 113697; Bernardi, A.V., Int. J. Mol. Sci. 2021, 22, 276; Bernardi, A.V., Int. J. Mol. Sci. 2019, 20, 2261, and other groups such as Wang et al. Biotechnol Biofuels (2019) 12:48
in lines 322-323, the authors say “Except glucose, we detected no sugar, which indicated that transglycosylation did not occur when only glucose was present”, yet lanes 3-5 show spots with variable Rf values on the TLC plate that are different to lane 1 (glucose alone). Can the authors revise the result narrative and discussion in this regard?
We are truly grateful for your critical comments. We believe that this change in the Rf of the samples compared to the glucose standard occurred due to a small difference in the running of the mobile phase on the silica plate, which normally occurs at the edges of the plate. Despite this small variation, it is evident that no product other than glucose appeared, which shows that the enzyme is not capable of forming transglycosylation in the presence of glucose alone. This result is in agreement with other works in the literature that have characterized beta-glycosidases with transglycosylation activity. Chang, Jie, et al. "Cloning, expression, and characterization of β-glucosidase from Exiguobacterium sp. DAU5 and transglycosylation activity." Biotechnology and Bioprocess Engineering 16 (2011): 97-106.; Kang, Liqin, et al. "β-Glucosidase BGL1 from Coprinopsis cinerea Exhibits a Distinctive Hydrolysis and Transglycosylation Activity for Application in the Production of 3-O-β-d-Gentiobiosyl-d-laminarioligosaccharides." Journal of agricultural and food chemistry 67.38 (2019): 10744-10755.; Zhao, Jun, et al. "Identification of an intracellular β-glucosidase in Aspergillus niger with transglycosylation activity." Applied Microbiology and Biotechnology 104 (2020): 8367-8380.; Uchiyama, Taku, Kentaro Miyazaki, and Katsuro Yaoi. "Characterization of a novel β-glucosidase from a compost microbial metagenome with strong transglycosylation activity." Journal of Biological Chemistry 288.25 (2013): 18325-18334.
In lanes 335-336, the authors say “when the enzyme is incubated with cellobiose only, the main product of the reaction is glucose, and no transglycosylation product is detected.”, yet, lanes 3 and 7 in Fig7A clearly shows that cellobiose produced both glucose and cellotriose in the presence of the enzyme. Can the authors revise the result narrative and discussion in this regard?
Reply: Thank you for your comments. We have modified the sentence and hope it is better: “However, when Td2F2 is incubated with cellobiose only, the main product of the reaction is glucose, and no transglycosylation product is detected. This contrasts with AfBgl1.3, which can form a product in the presence of cellobiose without glucose addition (Figure 7A-lanes 3 and 7 [78]”.
The authors tested the commercial cellulase cocktail, Celluclast 1.5L, known to be concentrated with higher cellobiohydrolase activity, particularly Cel7A, on CMC, a substrate meant to test endo-glucanase activity. This is not the best choice of the substrate to test for the complete cocktail’s suite of activities, substrates such as Avicel PH101 or Bacterial crystalline cellulose would have sufficed for this. It is also known that native cellulose is rich in crystalline regions than amorphous regions, therefore, results from CMC cannot be translated into a real-world context. In this regard, the experimental setup for the enzyme synergy studies is flawed.
Reply: We are truly grateful for your critical comments. However, our intention was to show the effect of AfBgl1.3 when added in Celluclast 1.5L using simple substrates and not complex biomasses. In fact, Avicel PH101 or Bacterial crystalline cellulose would be a better choice, but Brazil is suffering from delays in reagent delivery, and we did not receive it in time to present the results. Furthermore, Gundupalli et al, 2021 showed CMC hydrolysis using Celluclast 1.5L and determined the production of reducing sugars in the presence and absence of salts. They calculated kinetic parameters for Celluclast in the presence of CMC and showed the values Vmax 0.940 mg/mL min and Km 0.667 mg/mL (Bioprocess and Biosystems Engineering (2021) 44:2331-2344). Thus, we believe that the experimental setup employed is correct and allows us to conclude that the condition of increased hydrolysis activity.
Also, Celluclast 1.5L contains minimal glucosidase activity, hence the suppliers recommend using it with Novozyme 188, a glucosidase preparation, to achieve the conversion of cellooligosaccharides to glucose. Therefore, the addition of the glycosidase in this study may not necessarily be synergistic but merely additive. To confirm this, the authors should monitor glucose release by Celluclast 1.5L in the presence and absence of glycosidase.
Replay: We totally agree. We´ve rewritten the sentence.
Title:
The title should be revised to “Characterization of a new glucose-tolerant GH1 β-glycosidase from Aspergillus fumigatus with transglycosylation activity”
Replay: Done
Abstract:
Line 19, change “stimulation” to “co-incubation”
Replay: Done
Line 27, change “in synergy” to “synergistically”
Replay: Done
Introduction:
Line 46-49, rewrite as “reducing end-acting cellobiohydrolases (EC 3.2.1.176) and non-reducing end-acting cellobiohydrolases (EC 3.2.1.91) release cellooligosaccharide units like cellobiose”
Replay: Done
Results and discussion:
Line 100, rewrite as “Multiple sequence alignment”
Replay: Done
Line 107-109, describe “X” as a variable amino acid in the sequence motifs for the reader
Replay: Done
Line 117, rewrite as “Expression of recombinant β-glycosidase, AfBgl1.3, in P. pastoris and its characterization”
Replay: Done
Line 124, rewrite as “GH1 β-glycosidases”
Replay: Done
Line 147, change “submitting” to “subjecting”
Replay: Done
Line 153-154, the GH family affiliation comes before the enzyme’s name
Replay: Done
Line 220, change “including” to “such as”
Replay: Done
Line 365, fix “released sugars release”
Replay: Done
Line 629, rewrite as “for CMC and SDB saccharification”
Replay: Done

Reviewer 2 Report
The manuscript " Characterization of a new glucose-tolerant β-glycosidase GH1 2 from Aspergillus fumigatus with transglycosylation activity " is a novel idea to consider in the field
for new cellulases. The comments as follows:
1: Figure1, Figure3-6 are not clear.
2: The discussion is insufficient.
Author Response
The manuscript " Characterization of a new glucose-tolerant β-glycosidase GH1 2 from Aspergillus fumigatus with transglycosylation activity " is a novel idea to consider in the field for new cellulases. The comments as follows:
1: Figure1, Figure3-6 are not clear.
Reply: We are truly grateful for your critical comments. We´ve improved the quality of figures.
2: The discussion is insufficient.
Reply: Reply: We are truly grateful for your critical comments. We have added some comments in the discussion, as suggested by the other reviewers..
Reviewer 3 Report
Due to the fact that enzymatic digestion of the plant debris still remains a challenge novel enzymes are sought. The manuscript presents comprehensive and well-designed study focusing on identification and description of a novel enzyme β-glycosidase GH1 2 from A. fumigatus. The authors identified a gene which encodes the protein, described physicochemical properties of the enzyme, cloned it within yeast model and compared its activities with known, similar enzymes.
The overall quality of the manuscript is very high thus I have no questions regarding its presentation.
Author Response
Due to the fact that enzymatic digestion of the plant debris still remains a challenge novel enzymes are sought. The manuscript presents comprehensive and well-designed study focusing on identification and description of a novel enzyme β-glycosidase GH1 from A. fumigatus. The authors identified a gene which encodes the protein, described physicochemical properties of the enzyme, cloned it within yeast model and compared its activities with known, similar enzymes.
The overall quality of the manuscript is very high thus I have no questions regarding its presentation.
Reply: We are truly grateful for your critical comments. Thank you very much for your review of our manuscript (ijms-2149764).
Reviewer 4 Report
This is a well written paper describing a very thorough characterisation of this novel glycosidase enzyme. The results will be of particular interest to industrial chemists in the biorefinery industry.
I make only two small comments:
Line 186: I don't think that an enzyme that uses lactose as a substrate can be described as a cellobiase.
line 283 - I think this statement could be phrased more concisely without repeating "glucose": Another factor that prevents enzymes from being applied in the industry is their inhibition by glucose—glucose at high concentrations decreases the efficiency of enzymatic hydrolysis
Author Response
This is a well written paper describing a very thorough characterisation of this novel glycosidase enzyme. The results will be of particular interest to industrial chemists in the biorefinery industry.
Reply: We are truly grateful for your critical comments. Thank you very much for your review of our manuscript (ijms-2149764).
I make only two small comments:
Line 186: I don't think that an enzyme that uses lactose as a substrate can be described as a cellobiase.
Reply: We are in complete agreement. We´ve removed the lactose from the text.
line 283 - I think this statement could be phrased more concisely without repeating "glucose": Another factor that prevents enzymes from being applied in the industry is their inhibition by glucose—glucose at high concentrations decreases the efficiency of enzymatic hydrolysis
Reply: We´ve rewritten the sentence. ."Another factor that prevents enzymes from being applied in the industry is their inhibition by glucose, which at high concentrations decreases the efficiency of enzymatic hydrolysis. Therefore, β-glycosidases tolerant and stimulated by glucose are desirable for enzyme cocktails because they can increase the concentration of fermentable sugars at the end of the process "

Round 2
Reviewer 1 Report
N/A